# Labeled oxytocin administered via the intranasal route reaches the brain in rhesus macaques

M. R. Lee [1,8✉], T. A. Shnitko[2,8], S. W. Blue[2], A. V. Kaucher [3], A. J. Winchell[3], D. W. Erikson [3], K. A. Grant [2,4,9] & L. Leggio[1,5,6,7,9]

Oxytocin may have promise as a treatment for neuropsychiatric disorders. Its therapeutic effect may depend on its ability to enter the brain and bind to the oxytocin receptor. To date, the brain tissue penetrance of intranasal oxytocin has not been demonstrated. In this non-human primate study, we administer deuterated oxytocin intranasally and intravenously to rhesus macaques and measure, with mass spectrometry, concentrations of labeled (exogenously administered) and endogenous oxytocin in 12 brain regions two hours after oxytocin administration. Labeled oxytocin is quantified after intranasal (not intravenous) administration in brain regions (orbitofrontal cortex, striatum, brainstem, and thalamus) that lie in the trajectories of the olfactory and trigeminal nerves. These results suggest that intranasal administration bypasses the blood–brain barrier, delivering oxytocin to specific brain regions, such as the striatum, where oxytocin acts to impact motivated behaviors. Further, high concentrations of endogenous oxytocin are in regions that overlap with projection fields of oxytocinergic neurons.

[1] Section on Clinical Psychoneuroendocrinology and Neuropsychopharmacology, National Institute on Alcohol Abuse and Alcoholism, Division of Intramural Clinical and Biological Research and National Institute on Drug Abuse Intramural Research Program, National Institutes of Health, 10 Center Drive (10CRC/ 15330), Bethesda, MD, USA. [2] Division of Neuroscience, Oregon National Primate Research Center, Oregon Health & Science University, 505 NW 185th Avenue, Beaverton, OR, USA. [3] Endocrine Technologies Core, Oregon National Primate Research Center, Oregon Health & Science University, 505 NW 185th Avenue, Beaverton, OR, USA. [4] Department of Behavioral Neuroscience, Oregon Health & Science University, 3181 SW Sam Jackson Park Road, L-470, Portland, OR, USA. [5] Medication Development Program, National Institute on Drug Abuse Intramural Research Program, National Institutes of Health, Baltimore, MD, USA. [6] Center on Compulsive Behaviors, National Institutes of Health, Bethesda, MD, USA. [7] Center for Alcohol and Addiction Studies, Department of Behavioral and Social Sciences, Brown University, Providence, RI, USA. [8] These authors contributed equally: Lee M.R., Shnitko T.A. [9] These authors jointly supervised this work: Grant K.A., Leggio L. ✉email: leemary@mail.nih.gov

The nine amino acid peptide oxytocin (OT) may represent a novel treatment for neuropsychiatric disorders such as autism[1], mood disorders[2], and addiction[3,4]; the effect of exogenous OT to modulate symptoms related to these disorders is thought to be centrally mediated, with OT acting at its receptor in the brain[2,5]. Given this proposed central effect, evidence of the location and extent of brain parenchyma penetrance of OT after intranasal (IN) or intravenous (IV) administration is important to establish, in order to interpret results of clinical studies with OT.

The work presented here follows our previous study[6] where: (1) we developed a liquid chromatography-mass spectrometry-tandem mass spectrometry (LC–MS/MS) assay that accurately measured endogenous OT (d0OT) and administered OT (d5OT) and (2) we demonstrated that deuterated OT, d5OT, reaches the cerebrospinal fluid (CSF) in monkeys when administered by IN and IV routes. In that study[6], we sampled blood and CSF up to 1 hour (h) after IN and IV d5OT administration of a single 80 IU dose. At 1 h post IN and IV administration, levels of d5OT in the CSF were present and significantly elevated above baseline (i.e., significantly different from 0). Endogenous CSF levels of OT over 1 h were unaffected by d5OT administration via either route, indicating no evidence of a feed-forward effect on endogenous OT, at least within the timeframe of that study[6]. The pharmacokinetic parameters of d5OT in the CSF indicated that there was also no evidence of a privileged pathway to the CSF via the IN route.

Beyond CSF penetrance, the actual therapeutic effect of OT may depend on its ability to enter the brain and bind to the OT receptor. To date, the brain tissue penetrance of OT when delivered by the clinically relevant IN route has not been demonstrated. Therefore, in this nonhuman primate study, we administered d5OT intranasally to rhesus macaques and measured, with a LC–MS/MS assay, concentrations of labeled, as well as endogenous, OT in 12 brain regions. In this way, we investigated the location and extent of brain tissue penetration after IN and IV administration of d5OT. The 12 brain regions: cerebellum, brainstem, striatum, amygdala, thalamus, visual cortex, insular cortex, hypothalamus, hippocampus, medial prefrontal cortex (mPFC) (Brodmann areas 32, 25, 24), dorsal PFC (dPFC) (Brodmann areas 46, 9, 10), and orbital frontal cortex (OFC) (Brodmann areas 14, 13, 12) were chosen, as they are important in mediating social cognition in primate species[7,8]. Additionally, these regions are associated with the mesolimbic social decision-making network in primate species[9].

Brain concentrations were measured approximately 2 h after d5OT administration under three conditions: 40 IU IN, 80 IU IN, or 80 IU IV. We included the 40 IU IN dose in this study because (1) IN was the route of administration used in clinical studies with OT, and (2) 40 IU IN was in the upper range of doses commonly used in human studies with OT (20–40 IU)[10]. Further, measuring d5OT brain concentrations >2 h after administration, when CSF d5OT concentrations had been eliminated, may reveal information on OT receptor location.

Prior to the terminal study, using a within-subject design, we also attempted to determine the pharmacokinetics of d5OT and d0OT in the plasma and CSF after dosing with d5OT in 4 dose–route combinations: 40 IU and 80 IU delivered IN and IV over an extended time course, i.e., 2 h. Measurement of d5OT and d0OT in plasma, CSF, and brain was conducted with an LC–MS/MS assay that was developed based on our previous assay reported in:[6].

While our previous study did not indicate that IN delivery of d5OT was a privileged pathway to the CSF, there remains the question of whether IN delivery of OT is a delivery method that bypasses the blood–brain barrier. By employing IV administration of d5OT as a comparison condition, we were able to address this question. Further, as numerous elegant rodent studies using viral vectors and optogenetics report central projections from hypothalamic oxytocinergic cell bodies to diverse brain regions[11–14], our approach in the present study allowed us to measure brain concentrations of endogenous OT across these regions in a primate species.

We quantified labeled OT after IN (not IV) administration in brain regions (orbitofrontal cortex, striatum, brainstem, and thalamus) that lie in the trajectories of the olfactory and trigeminal nerves. Quantification after IN, in contrast to IV delivery, suggests that IN administration bypasses the blood–brain barrier, delivering OT to specific brain regions, such as the striatum, where OT acts to impact motivated behaviors. High concentrations of endogenous OT are in regions that overlap with projection fields of oxytocinergic neurons.

## Results

**Brain concentrations of d5OT and d0OT.** Concentrations of d5OT were analyzed in all 12 brain regions (see Table 1). D5OT was quantified only after IN administration, not IV administration; the brain regions where d5OT was quantified were OFC, striatum, thalamus, and brainstem. In addition, d5OT was detected, but not quantified, in cerebellum, amygdala, dlPFC, and mPFC. Table 1 tabulates the brain tissue d5OT concentration (pg/mg) with the corresponding nM concentration for each DOSE–ROUTE condition. Endogenous (d0) OT brain tissue concentrations were quantified in all 12 brain regions, with the highest concentration in hypothalamus, followed by amygdala, brainstem, striatum, hippocampus, and OFC. Of note, the lower concentrations were in the visual and insular cortices, as well as the medial and dorsal prefrontal cortices, compared with the aforementioned regions. These are tabulated in Table 2 (pg/mg) and Table 3 (nM).

**CSF concentrations of d5OT.** There was no d5OT detected in CSF at $T = 0$, as expected. D5OT in CSF was detected at timepoint 60, after 40 IU IV and 80 IU IV d5OT administration; there was no d5OT detected at 120 min after IV administration (Fig. 1). After IN administration of either dose of d5OT, there was no d5OT detected in the CSF at any timepoint. D5OT was detected just prior to necropsy, after 80 IU IV in a single monkey, but was below the limit of quantification for the assay. Planned statistical tests were not conducted for CSF d5OT concentrations, as no single concentration was equal to, or above, the limit of quantification of the assay used in this study (100 pg/ml).

**Plasma concentrations of d5OT.** There was no d5OT detected in plasma at $T = 0$. All concentrations of d5OT in the plasma after 40 IU IN administration were below the level of quantification for the LC–MS/MS assay used in this study. Time courses for plasma d5OT for each of the four DOSE–ROUTE combinations are shown in Fig. 2. Plasma d5OT concentrations (above or equal to the limit of quantification) were available for the 80 IU IN, 80 IU IV, and 40 IV conditions, however, data were not normally distributed and were unable to be normalized. Therefore, planned statistical tests were not conducted and, instead, exploratory pairwise comparisons were conducted with the Wilcoxon Signed Ranks Test to (1) compare concentrations across DOSE-ROUTE conditions at each timepoint (60 and 120 min) and (2) compare concentrations within each DOSE–ROUTE condition across timepoints. At each timepoint (60 and 120 min), there was no significant difference ($p = 0.07$ and 0.59, respectively) between d5OT plasma concentrations in the 80 IU IN compared with 80 IU IV condition, indicating that one route of administration of a

**Table 1 Concentrations of administered deuterated oxytocin (d5OT) in the brain of monkeys.**

| Dose–Route d5OT | 80 IU IN | | 40 IU IN | | 80 IU IV | |
|---|---|---|---|---|---|---|
| Monkey ID# (sex) | ID #2 (M) | ID #4 (M) | ID #1 (F) | ID #5 (F) | ID #3 (F) | ID #6 (F) |
| Brain region | BRAIN d5OT concentration pg/mg (nM) | | | | | |
| Cerebellum | ND | + | ND | ND | ND | ND |
| Brainstem | 0.14 (<1) | 0.24 (1) | + | ND | ND | + |
| Striatum | ND | ND | 2.41 (13) | + | ND | ND |
| Amygdala | ND | ND | NM | + | ND | ND |
| Thalamus | ND | ND | 0.23 (1) | ND | ND | ND |
| Visual cortex | ND | ND | ND | ND | ND | ND |
| Insular cortex | ND | ND | NM | ND | ND | ND |
| dPFC | ND | + | ND | + | ND | + |
| mPFC | ND | ND | ND | ND | ND | + |
| OFC | 2.04 (11) | ND | ND | ND | ND | ND |
| Hypothalamus | ND | | ND | | ND | |
| Hippocampus | ND | | ND | | ND | |

*ND* not detected, *NM* not measured, + detected but not quantifiable, *OT* oxytocin, *dPFC* dorsal prefrontal cortex, *mPFC* medial prefrontal cortex, *OFC* orbital frontal cortex, *CSF* cerebrospinal fluid, *F* female, *M* male, *IN* intranasal, *IV* intravenous.

**Table 2 Concentrations (pg/mg) of endogenous brain oxytocin (OT).**

| Dose–Route d5OT | 80 IU IN | | 40 IU IN | | 80 IU IV | |
|---|---|---|---|---|---|---|
| Monkey ID# (Sex) | ID #2 (M) | ID #4 (M) | ID #1 (F) | ID #5 (F) | ID #3 (F) | ID #6 (F) |
| Brain Region | BRAIN OT concentration (pg/mg) | | | | | |
| Cerebellum | ND | 0.06 | 2.16 | 0.12 | ND | + |
| Brainstem | 5.11 | 3.55 | 7.27 | 9.34 | 10.93 | 3.58 |
| Striatum | 1.58 | 17.81 | 0.63 | 1.20 | 14.45 | 15.82 |
| Amygdala | 5.82 | 23.97 | NM | 39.11 | 4.63 | 7.26 |
| Thalamus | 1.02 | 3.09 | 4.39 | 7.31 | 3.85 | 28.34 |
| Visual Cortex | ND | ND | ND | ND | ND | ND |
| Insular Cortex | ND | 0.06 | | 0.30 | 0.16 | ND |
| dPFC | ND | 0.20 | ND | 0.16 | 0.27 | ND |
| mPFC | ND | 0.45 | 0.86 | 0.88 | ND | ND |
| OFC | 0.40 | 1.59 | 0.22 | 0.96 | 0.44 | ND |
| Hypothalamus | 943.27 | | 2186.74 | | 1491.96 | |
| Hippocampus | 2.83 | | 0.99 | | 0.99 | |

*ND* not detected, *NM* not measured, + detected but not quantifiable, *OT* oxytocin, *dPFC* dorsal prefrontal cortex, *mPFC* medial prefrontal cortex, *OFC* orbital frontal cortex, *CSF* cerebrospinal fluid, *M* male, *F* female, *IN* intranasal, *IV* intravenous.

given dose did not result in significantly higher plasma d5OT concentrations compared to the other. Administration of 80 IU IV compared with 40 IU IV resulted in significantly, and dose-dependently, higher d5OT plasma concentrations at 60 and 120 min, $p = 0.03$ and 0.04, respectively. There was no significant difference between plasma concentrations of d5OT at 60 min, compared with 120 min, in the 80 IU IN condition ($p = 1.00$), indicating that plasma concentrations of d5OT were sustained over 2 h after IN administration. In contrast, there was a significant reduction in d5OT concentration from 60 to 120 min in the 80 IU IV ($p = 0.04$) and in the 40 IU IV ($p = 0.03$) conditions, as expected. At necropsy, plasma d5OT was quantified only in the 80 IU IV condition, where concentrations were 271 and 499 pg/ml for monkey #3 and #6, respectively.

**Endogenous d0OT.** D0OT in plasma was not detected. Figure 3 shows the time course of d0OT concentrations for each monkey in the CSF at $t = 0$, 60, and 120 min for each DOSE-ROUTE

combination. CSF d0OT concentrations above or equal to the limit of quantification (10 pg/ml) were normalized with a ln transformation (K–S: $p = 0.20$). There was no significant difference in baseline ($T = 0$) ln (d0OT) concentrations in CSF [F-(3,10.08) = 2.14, $p = 0.16$]. There was a main effect of TIME [F (2,2.02) = 57.12, $p < 0.02$] (Fig. 3), with a reduction of ln (d0OT) concentrations over the 2 h time course of CSF sampling. There was no main effect of DOSE–ROUTE condition [F(3,6.01) = 0.80, $p = 0.54$]. There was no DOSE–ROUTE × TIME interaction [F(6,3.83) = 0.93, $p = 0.56$]. CSF concentrations of d0OT at necropsy in the 80 IU IN condition were 16 and 10 pg/ml, in the 40 IU IN condition 12 pg/ml (not quantified in monkey #1), and in the 80 IU IV condition 16 and 14 pg/ml.

Bivariate correlation between d0OT concentration in plasma and CSF was not possible as endogenous OT in plasma was not able to be quantified. Bivariate correlation between d0OT in the CSF and d5OT concentration in the plasma was not significant: Pearson correlation = 0.07, $p = 0.655$.

**Table 3 Concentrations (nM) of endogenous brain oxytocin (OT).**

| Monkey ID # (Sex) | 80 IN | | 40 IN | | 80 IV | |
|---|---|---|---|---|---|---|
| | ID #2 (M) | ID #4 (M) | ID #1 (F) | ID #5 (F) | ID #3 (F) | ID #6 (F) |
| Cerebellum | ND | <1 | 12 | <1 | ND | + |
| Brainstem | 28 | 19 | 40 | 51 | 60 | 20 |
| Striatum | 9 | 97 | 3 | 7 | 79 | 86 |
| Amygdala | 32 | 131 | NM | 214 | 25 | 40 |
| Thalamus | 6 | 17 | 24 | 40 | 21 | 155 |
| Visual Cortex | ND | ND | ND | ND | ND | ND |
| Insular Cortex | ND | <1 | NM | 2 | 1 | ND |
| dPFC | ND | 1 | ND | 1 | 2 | ND |
| mPFC | ND | <1 | <1 | <1 | ND | ND |
| OFC | 2.2 | 9 | 1 | 5 | 2 | ND |
| Hypothalamus | 368 | | 3200 | | 895 | |
| Hippocampus | 15 | | 5 | | 5 | |

*ND* not detected, *NM* not measured, + detected but not quantifiable, *OT* xytocin, *dPFC* dorsal prefrontal cortex, *mPFC* edial prefrontal cortex, *OFC* orbital frontal cortex, *CSF* cerebrospinal fluid, *M*male, *F* female, *IN* intranasal, *IV* intravenous.

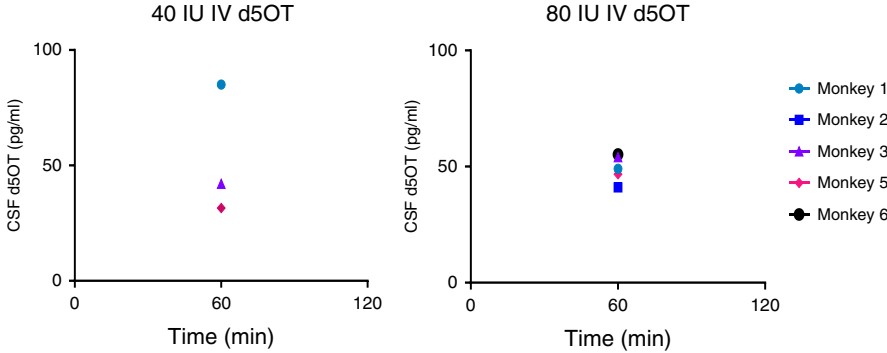

**Fig. 1 Cerebrospinal fluid (CSF) d5 oxytocin (OT) concentrations: time course of d5OT concentrations in monkey cerebrospinal fluid (CSF) after intravenous (IV) administration of 80 and 40 IU d5OT (N = 6).** No d5OT was detected after intranasal (IN) administration of 80 IU d5OT. Limit of quantification = 100 pg/ml.

## Discussion

This study extends our previous work[6], and others[15,16], by examining the location, extent, and mechanism of brain tissue penetrance of d5OT after IN administration of two d5OT doses, including 40 IU of d5OT, which is the clinically important route and dose used in a multitude of clinical studies[17]. Comparison with the IV condition indicates that the mechanism of brain penetrance for the IN condition involves bypassing the blood–brain barrier. To our knowledge, this is the first study to report that d5OT, administered IN in doses of 40 or 80 IU, reaches the brain with quantifiable concentrations in OFC, brainstem, striatum, and thalamus (Table 1). Notably, d5OT in the brain was detected 2 h after IN administration, when d5OT was not detectable in the CSF under these same conditions. In contrast, after IV administration, there was no quantifiable d5OT in any brain region in the two monkeys who received 80 IU IV d5OT. This difference indicates, as has been posited, that d5OT administered via the IN route passes into the brain directly, bypassing the blood–brain barrier[18].

The brain regions where d5OT was quantified are regions innervated by the olfactory (striatum, OFC, thalamus) and trigeminal (brainstem) nerves. Brain penetrance after IN delivery may occur by extracellular transport (diffusion along peripheral olfactory or trigeminal nerves), intracellular transport (axonal transport within olfactory/trigeminal nerves), or extracellular convection (bulk flow) along the olfactory or trigeminal nerves, which both innervate the nasal cavity. Quantifiable brain concentrations 2 h after IN administration suggest bulk flow as the most probable mechanism for transport along both olfactory and trigeminal nerves[19]. Once reaching the olfactory bulb and brainstem, movement of the peptide to further brain regions can also occur by intra- or extracellular processes[19]. It is also possible that, apart from these neuronal pathways, IN administration itself facilitates this transport into the brain by physical disruption of the blood–brain barrier. However, if this were the case, more widespread brain penetrance would have occurred.

In an IN administration study[20] using labeled interferon, the striatum was the site of the most robust accumulation of labeled interferon, after 60 min. As noted above, this time course favors an extracellular bulk flow mechanism for transport within perivascular spaces in the brain to distant brain regions, such as the striatum. We also report here that the striatum was the site of the highest concentration of d5OT, comparable to that measured in the OFC, ~10 nM (Table 1). In a recent postmortem analysis of human brain[21], the striatum had the highest concentration of OT receptor mRNA compared with any other brain region, although the OT receptor has not been found in the striatum of macaque species with competitive binding assays[22]. While different techniques and methodologies may account for this discrepancy, our findings support the possibility that the high striatal d5OT concentration after IN d5OT is related to OT receptor binding to

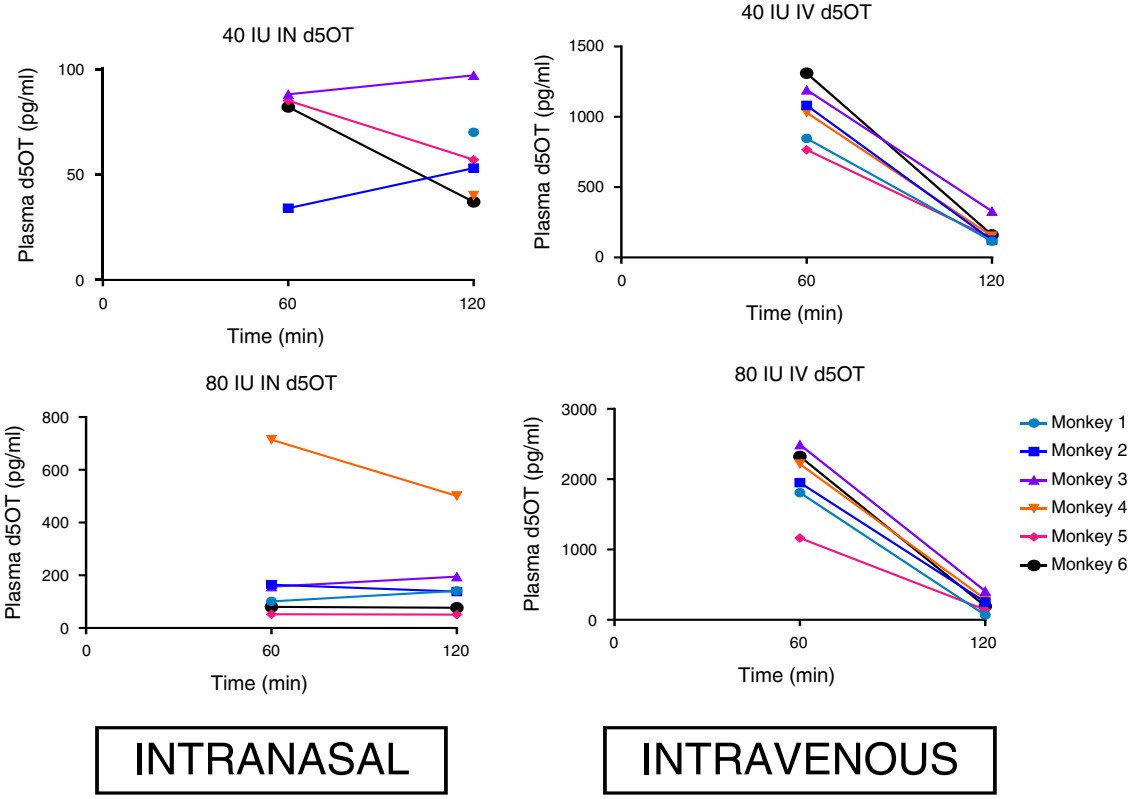

**Fig. 2 Plasma d5 oxytocin (OT) concentrations: time course of d5OT concentrations in monkey plasma after intravenous (IV) and intranasal (IN) administration of 80 and 40 IU d5OT (N = 6).** Limit of quantification = 100 pg/ml.

d5OT. Further, there is evidence from imaging studies that OT delivered IN alters striatal neural function. In a human study, OT delivered IN (40 IU) increased resting cerebral blood flow in striatum 1 h after administration compared to placebo[23]. Downstream, the same IN dose increased resting state functional connectivity between caudate and dorsal anterior cingulate ACC, a pathway implicated in motivated behaviors and reward learning[24]. In a human postmortem study using this competitive binding assay, significant OT receptor radioligand binding was measured in several midbrain structures[25]. That said, OT fibers are more restricted in the rhesus macaque brain compared with humans or chimpanzees[26].

The selective accumulation of d5OT in the striatum and other brain regions along the trajectory of the trigeminal and olfactory nerves, after IN administration in nonhuman primates, has clinical significance for translation to humans. First, the structure and function of the nasal cavity is similar between humans and monkeys[19,27]: the olfactory region is located in the upper nasal cavity and is not affected by inspired air; the cilia of the olfactory mucosa are not motile, and the olfactory region covers the same percentage of the surface area of the nasal cavity. Therefore, overall, the target site of the intranasally sprayed substance is similar between monkeys and humans. The mucosal atomization delivery device produces droplet sizes 30–100 μm; deposition of this droplet size results in delivery of ~20% of the delivered volume to the upper nasal cavity[28]. Taking this into account, brain penetrance of the 40 IU dose of d5OT in the striatum is on the order of 0.03% of the dose delivered IN. While the brain recovery in this region of the delivered dose is low, the brain concentrations of d5OT after IN delivery are on the order of 1 nM or greater (Table 1), where 1 nM is required for signaling at the OT receptor in extrahypothalamic brain regions[29]. Lastly, these physiologically relevant brain

concentrations in the striatum after IN administration are relevant for the putative effect of OT, as the striatum is the brain region where OT acts to modulate motivated behaviors[11,30–32].

The regional endogenous brain OT concentrations (Tables 2 and 3) are quite variable within each dose–route condition and across doses. The regions with consistently elevated d0OT concentrations (brainstem, striatum, amygdala, thalamus, and hypothalamus) overlap partially with regions (olfactory cortex, putamen, pallidum, and thalamus) that have higher than average OT mRNA expression in human brain[21]. Endogenous brain OT is thought to originate largely from hypothalamic dendritic release and subsequent diffusion to distant brain sites[33], rather than from local release from axon terminals. However, brain regions in the present study with elevated endogenous OT concentrations overlap with those regions where axon terminals of hypothalamic neurons have been found, namely the amygdala, hippocampus, striatum, and somatosensory cortex[11–14], suggesting that axon terminal release may play a significant role in contributing to regional endogenous brain OT concentrations.

As in our previous study[6], the results of the present study indicate that IN delivery does not provide a privileged pathway to the CSF as d5OT was not detected in the CSF after IN, but was detected after IV administration. The assay used here had a limit of quantification tenfold more than that reported in the previous study[6], where the limit of quantification was 10 pg/ml. Further, at 60 min in the previous study, the CSF concentrations of d5OT after IN delivery of 80 IU were highly variable (11, 16, 67, and 221 pg/ml), and would have been detectable in only two of the monkeys using the assay in the present study, which had an MDL of 38 pg/ml. Finally, further studies are needed during a shorter time window (<60 min) to investigate the variability of IN delivery.

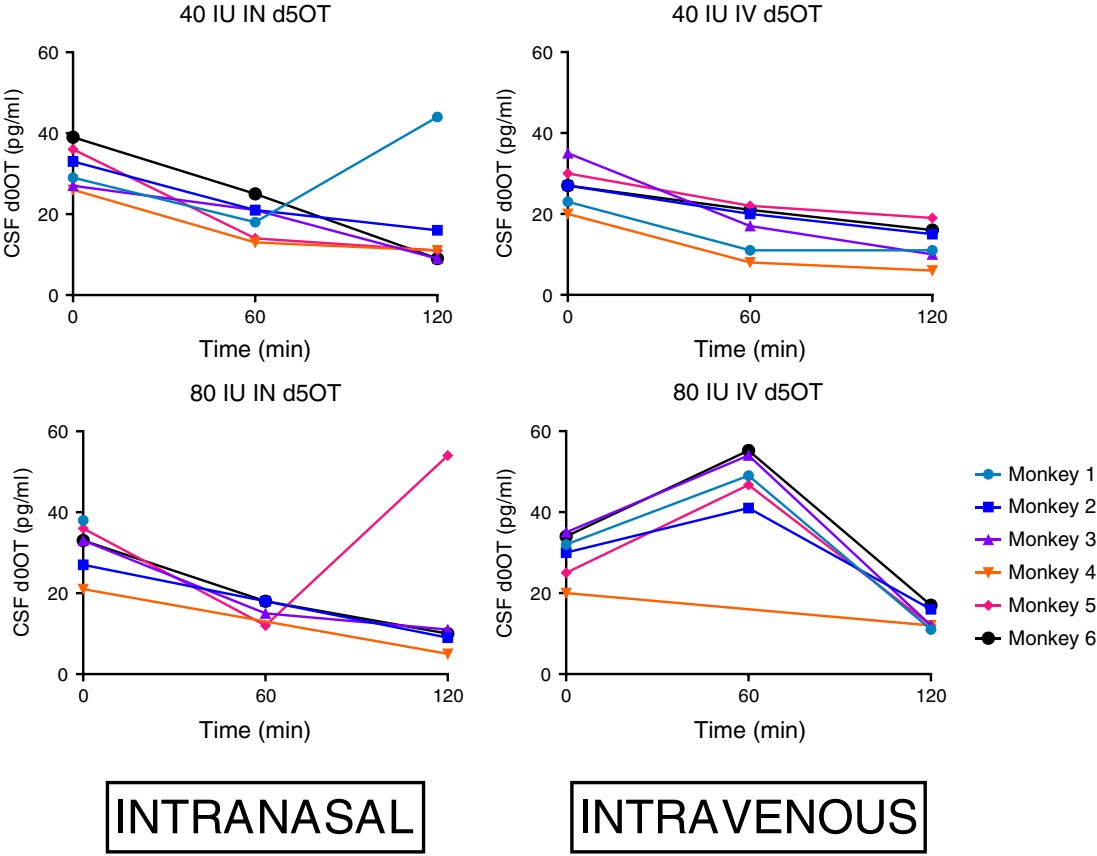

**Fig. 3 Cerebrospinal fluid (CSF) endogenous (d0) oxytocin (OT) concentrations: time course of d0OT in monkey cerebrospinal fluid (CSF) at baseline (Time = 0) and after intranasal (IN) d5OT 80 and 40 IU; after intravenous (IV) d5OT 80 and 40 IU.** Limit of quantification = 10 pg/ml.

For endogenous OT concentrations in CSF, there was no difference between the four dose–route conditions in the time course of d0OT CSF concentrations over 2 h. As we had no saline condition, we can only conclude that there was no differential effect of the four dose–route conditions on the time course of endogenous CSF OT. There was a main effect of time on d0OT CSF concentrations, where concentrations of d0OT decreased over time, regardless of OT administration, similar to our previous study[6]. Decreasing concentrations of d0OT in CSF, ~50% over 2 h, were observed in all four dose–route conditions. Therefore, these results do not support a feed-forward effect of administered d5OT on endogenous levels. If the latter were the case, we would have expected increasing concentrations of d0CSF over time, after d5OT administration. Further, although hypothalamic tissue concentrations of d5OT were not detected and endogenous OT is generally thought to stimulate its own release[33], we cannot exclude the possibility that d5OT administration led to a pharmacologically-induced negative feedback and suppressed endogenous CSF OT acting at the hypothalamus.

This study has several limitations. The limit of quantification (100 pg/ml) and between subject variability for d5OT precluded comparisons of CSF concentrations of d5OT across routes of administration (IN vs. IV), or across time (60 vs. 120 min), for a given route. Our previous study[6] reported greater sensitivity in the d5OT assay by LC–MS/MS, but both the previous assay and the assay used in the present study showed high variability. For example, mean plasma concentration of d5OT at 60 min in the IV condition was 1992 ± 476 pg/ml in the present study and 3306 ± 1162 pg/ml in the previous study. Plasma d5OT at 60 min in the

80 IU IN condition in the present study was 284 ± 288 pg/ml, compared with 615 ± 474 pg/ml in the previous study. This variability, as well as the non-normal distribution, may have contributed to the failure to detect a difference between plasma concentrations of d5OT after IN, compared with IV administration at each timepoint. In comparison, a difference was observed at 60 and 120 min in our previous study[6].

The long interval (2 h) between d5OT administration and necropsy is another limitation, as is the qualitative nature of the analysis of the brain concentrations (given only two monkeys per condition), and the considerable within- and across-condition variability, which precluded inferences about the effect of dose. Further studies, with a larger number of animals per condition and a shorter interval, may allow determination of regional brain distributions after each route of administration, and comparison between the routes and between doses. Through this comparison, we would be able to determine the extent to which IN administration results in brain concentrations of d5OT, in excess of that which crossed the blood–brain barrier (i.e., IV route). This determination is especially relevant, as a recent preclinical study reported a transport mechanism for OT across the blood–brain barrier[34]. Determining the extent and location of brain penetrance after clinically relevant IN administration is necessary, as the neurobehavioral effects of OT administered at peripheral sites, including IN, are most likely centrally mediated [reviewed in[4]], with OT signaling in the brain required for its behavioral effect[5].

There is the possibility that deuteration may have affected the pharmacokinetics of OT, as deuteration is used to stabilize and prolong the half-life of small molecules that are metabolized via

**Table 4 Subject Characteristics and Experimental Details.**

| Monkey | Sex | Age (years) | Weight (kg) | CSF and plasma sampling experiment | | | | | Necropsy experiment |
|---|---|---|---|---|---|---|---|---|---|
| | | | | DOSE–ROUTE condition for d5OT | | | | | Interval: d5OT admin. to perfusion in minutes |
| | | | | Week 1 | Week 2 | Week 3 | Week 4 | Week 5 | |
| 1 | F | 16.7 | 6.6 | 80 IU IV | 40 IU IV | 40 IU IN | 80 IU IN | 40 IU IN | 148 |
| 2 | M | 5 | 7 | 40 IU IV | 80 IU IN | 40 IU IN | 80 IU IV | 80 IU IN | 136 |
| 3 | F | 7.8 | 5.6 | 80 IU IN | 80 IU IV | 40 IU IV | 40 IU IN | 80 IU IV | 149 |
| 4 | M | 17.9 | 12 | 80 IU IV | 40 IU IN | 80 IU IN | 40 IU IV | 80 IU IN | 135 |
| 5 | F | 16 | 5.05 | 40 IU IN | 40 IU IV | 80 IU IV | 80 IU IN | 40 IU IN | 138 |
| 6 | F | 7.1 | 5.35 | 40 IU IN | 80 IU IN | 40 IU IV | 80 IU IV | 80 IU IV | 135 |

*CSF cerebrospinal fluid, OT oxytocin, IN intranasal, IV intravenous, F female, M male.*

oxidation[35]. However, this is unlikely as OT metabolism is via hydrolysis, therefore, the peptide's pharmacokinetic profile is unlikely to altered by deuteration. Lastly, the sample size of six monkeys in this terminal experiment was not adequate to detect sex differences in endogenous OT brain concentrations, or to calculate the effect of exogenous administration on endogenous brain OT concentrations. The brain d5OT concentrations after IN administration, as well as endogenous OT concentrations, were, in some regions, measured at biologically relevant concentrations (i.e., ≥1 nM). These results advance knowledge of OT brain penetrance as a function of route, i.e., IN administration. While IN delivery of OT is not a privileged pathway to the CSF, it is a privileged path to the brain, where it bypasses the blood–brain barrier and blood–CSF barrier. Via this route, OT can reach biologically active concentrations in regions that are relevant for OT to exert its neuromodulatory effect on social behaviors, addictive and compulsive disorders, and anxiety. The mechanism by which IN OT gains access to the brain should be further examined, similar to recent efforts with IN administration of PET tracers for manganese[36]. These privileged pathways to the brain may be leveraged to improve regional specificity of drug delivery to the brain, and avoid unnecessary systemic exposure of drugs aimed to target brain neurocircuitry.

## Methods

**Study setting**. This study was conducted at the Oregon National Primate Research Center (ONPRC). All procedures adhered to the NIH Guide for the Care and Use of Laboratory Animals and were approved by the ONPRC Institutional Animal Care and Use Committee. In the pharmacokinetic experiment, conducted over 4 consecutive weeks, rhesus macaques ($N = 6$) were dosed with four DOSE–ROUTE combinations of d5OT: 40 IU IN, 40 IU IV, 80 IU IN, or 80 IU IV. This dosing regime was followed by the terminal experiment in week 5, where three DOSE–ROUTE conditions of d5OT were administered (two monkeys per condition): 40 IU IN, 80 IU IN, or 80 IU IV. See Table 4 for details of the study timeline.

**Subjects**. Subjects were rhesus macaques (*Macaca mulatta*, $N = 6$, four females), weighing 5.05–12 kg, and were 5–18 years old. See Table 4 for subject characteristics and details of the study timeline.

**Oxytocin administration**. D5OT (Sigma-Aldrich, item #11799) 80 IU was dissolved in 1 ml sterile, normal saline, and sterile filtered before administration. For IV administration, 1 ml was administered via a peripheral IV catheter. For IN administration, an IN Mucosal Atomization Device [MAD, Teleflex Medical, Research Triangle Park, NC] was used. This same device was used in our previous nonhuman primate study[6]. It was also used, along with a nebulizer condition, to administer OT (48 IU) in another nonhuman primate study; both methods of delivery resulted in significant elevations in CSF OT concentrations[37]. Of note, another nonhuman primate study used a different IN device and reported no increased concentration of CSF OT after IN administration of unlabeled OT[38].

The MAD delivery device was attached to a 1 ml syringe; 0.5 ml of the d5OT solution was drawn up into the syringe. For the 40 IU IN condition, 0.5 ml was administered via one nostril. For the 80 IU IN condition, 0.5 ml was administered to each nostril. For dosing with the MAD, the monkeys were placed in a supine position, with head tilted back 60° below the horizontal plane. In a human cadaver study[39] using the MAD, this position resulted in optimal delivery of a similar volume of liquid to the ethmoid region.

**Pharmacokinetic analysis**. The six monkeys were dosed once a week, for 4 consecutive weeks, with each DOSE–ROUTE condition of d5OT (40 IU IN, 40 IU IV, 80 IU IN, or 80 IU IV). Each DOSE–ROUTE condition was assigned using a Latin square randomization (Table 4). Sessions were separated by at least 6 days. During each session, CSF (750 μl) and blood (5 ml) samples were drawn immediately prior to ($T = 0$) and following ($T = 60$ and 120 min) each administration of d5OT.

**Brain oxytocin concentration analysis**. Monkeys were divided into three groups, each with two monkeys; each group received the following in a terminal study at the 5th week: Group 1 (Monkeys ID# 1 and ID# 5) received 40 IU of d5OT IN, Group 2 received 80 IU of d5OT IN (Monkeys ID# 2 and ID# 4) and Group 3 received 80 IU of d5OT IV (Monkeys ID# 3 and ID# 6). Animals were perfused on average, $140 \pm 7$ (mean ± SD) min after d5OT administration (range = 135–149 min) (Table 4). Plasma and CSF were drawn just before administration of d5OT, at the midpoint between d5OT administration and perfusion (~60 min post-d5OT administration), and just before perfusion.

Brain tissue distribution of d5OT, as well as the endogenous concentrations of OT (d0OT), were measured in the following 12 brain regions: cerebellum, brainstem, striatum, amygdala, thalamus, visual cortex, insular cortex, hypothalamus, hippocampus, mPFC (Brodmann areas 32,25,24), dPFC (Brodmann areas 46,9,10), and OFC (Brodmann areas 14,13,12) (Tables 1, 2 and 3).

**Oxytocin assay**. *Chemicals*: An unlabeled standard for d0OT and deuterated standard for d5OT were purchased from Sigma-Aldrich (St. Louis, MO); d10OT was purchased for use as an internal standard through a custom order from Cayman Chemical (Ann Arbor, MI). All other reagents were commercially available, and we purchased LC or MS grade chemicals when available. Plasma and CSF sample preparation for LC–MS/MS analysis. CSF and plasma samples were collected as described above. Using 15 ml polystyrene tubes, 1.0 ml of each standard, quality control (QC), or plasma sample was combined with 30 μl of internal standard mixture (34 ng/ml d10OT in 50:50 acetonitrile (ACN):water, v–v) and incubated for 20 min on ice. Volumes used for CSF samples ranged from 0.4 to 1.0 ml, depending on sample availability. CSF samples <1.0 ml were brought to a final volume of 1.0 ml using charcoal-stripped serum, prior to addition of internal standard. To each tube, 2 ml of ice-cold ACN was added, vortexed thoroughly, and centrifuged at $2500 \times g$ for 10 min at 4 °C. The supernatants were transferred to new 13 mm × 100 mm polypropylene tubes and evaporated in a 37 °C water bath, under forced air, to a volume of <1 ml. We then added 5.5 ml of 0.2 M ammonium acetate (pH 3.0) in water, capped the tubes and mixed by inversion, before centrifugation at $2500 \times g$ for 10 min at 4 °C.

Solid-phase extraction (SPE) columns (Strata-X-Drug B, 60 mg/6 ml, Phenomenex, Torrance, CA) were conditioned prior to use by adding 2 ml of methanol, followed by 2 ml of water, and 2 ml of 0.2 M ammonium acetate (pH 3.0) in water; each was allowed to flow through by gravity before the next was added. The supernatants from the previous step were then added to SPE columns (one supernatant per column) by pouring. Columns were washed with 2 ml ammonium acetate (pH 3.0) in water, followed by 2 ml water and 2 ml ACN, and dried for 5 min in a vacuum manifold (5 in Hg; Millipore, Burlington, MA). OT was eluted with 2 × 1 ml 0.2% ammonium hydroxide in methanol into 12 mm × 75 mm polypropylene tubes and evaporated to dryness under forced air in a 37 °C water bath. The extract in each tube was reconstituted in 100 μl 0.1% formic acid in 5:95 ACN:water (v:v), vortexed thoroughly and transferred to 96-well filter plates (Millipore GV Multiscreen 0.22 μm Durapore). After a 15 min incubation at 4 °C, plates were filtered into microtiter plates (Shimadzu, Kyoto, Japan) using a positive

pressure manifold (8 min at 5 psi, followed by 15 s at 15 psi; Biotage, Uppsala, Sweden), and analyzed.

*Brain tissue sample preparation*: Samples (~1.5 mg per sample) were individually weighed and placed into 12 mm × 75 mm glass tubes containing 1.5 ml ice-cold 1 M acetic acid and 30 μl of OT-d10 internal standard. Tissues were homogenized on ice and the resulting homogenates were poured into individual 15 ml polystyrene tubes. The original glass tube was rinsed with 1 ml acetonitrile, and this rinse was added to the appropriate 15 ml polystyrene tube containing the homogenate. After all homogenizations were complete, 3 ml acetonitrile was added to each tube and extraction proceeded, as described for the plasma and CSF samples.

*Preparation of calibration curve and quality control samples*: D0OT and d5OT purchased from Sigma were prepared in 50:50 ACN:water and 1 ml aliquots of 100 μg/ml were stored at −80 °C. Calibration curves were prepared by thawing an aliquot and diluting further in 50:50 ACN, to working concentrations, for preparing the standard curve. We added 40 μl d0OT and 40 μl d5OT in 50:50 ACN:water (v/v) to each tube of 1 ml charcoal-stripped human serum (Golden West Biologicals, Temcula, CA). The final 11-point calibration curve ranged from 10 pg/ml to 80 ng/ml and included calibrators at 10 pg/ml, 25 pg/ml, 50 pg/ml, 100 pg/ml, 240 pg/ml, 720 pg/ml, 2 ng/ml, 6 ng/ml, 20 ng/ml, and 80 ng/ml. Calibrators were prepared immediately before sample preparation. QC samples were prepared by spiking unlabeled OT standard into normal human serum and stored at −80 °C. These QCs were assayed in triplicate at the front of each assay, and in duplicate after every 30 samples, and were accurate to within 85–115% of expected values.

*LC–MS/MS instrument parameters*: Microtiter plates were prepared, as described above, and loaded onto a SIL-30ACMP autosampler set at 4 °C (Shimadzu). Forty microliters of each standard, QC or sample were injected onto an ACE Excel 2 C18-PFP 50 mm × 2.1 mm column (Advanced Chromatography Technologies, LTD, Aberdeen, Scotland) at 50 °C using reversed-phase chromatography. Mobile phase A was 0.1% formic acid in water and mobile phase B was 0.1% formic acid in ACN. The LC time gradient was created using two Nexera LC-30AD pumps (Shimadzu) running at 0.5 ml/min. Chromatography for serum and CSF samples began at 10% B and ended at 16% B at 3.6 min. One minute at 60% B was then required to wash the column, and the pumps re-equilibrated at 10% B for an additional 1.7 min (total run time ~6 min). Outside of the 1.55–3.55 min window during which OT eluted, a divert valve sent the LC effluent to waste.

Heated electrospray injection in positive mode, with scheduled multiple reaction monitoring (MRM) on a Shimadzu LCMS-8050, was used for detection of OT. The interface temperature was 270 °C and heat block temperature was 170 °C. Gas was supplied by a Peak Genius 1051 nitrogen and air generator (Peak Scientific, Inchinnan, UK). Nitrogen gas was used for nebulizing (1 L/min) and drying (8.2 L/min), while air was used for heating (7.8 L/min). A capillary B needle was used in the interface. Needle protrusion was set to 1.25 mm and probe distance was set to 1.0 mm; both were determined empirically. Argon (Airgas, Radnor, PA) was used for the collision-induced dissociation at 330 kPa. The MS/MS conditions for OT were optimized using the automated MRM optimization procedure in LabSolutions software (Shimadzu). We collected two MRM transitions each, for d0OT (1007.40 > 723.40 and 1007.40 > 285.30), d5OT (1012.00 > 723.40 and 1012.00 > 290.35), and d10OT (1017.70 > 723.35 and 1017.70 > 295.40); the first transition listed in each case was used for quantitation, while the second was used as a reference ion. Retention times for d0OT, d5OT, and d10OT were 3.290, 3.270, and 3.253 min, respectively.

*Method validation*: The LC–MS/MS method was validated for analysis of d0OT and d5OT in plasma, CSF, and brain tissue by assessing recovery, matrix effects, specificity, precision, accuracy, stability, sensitivity, and reproducibility, largely by following the guidelines for analytical method validation set forth by the United States Food and Drug Administration[40].

The sensitivities of each assay within the method were determined by the lowest and highest concentration calibrators in the calibration curve. The lower limit of quantitation (LLOQ) for d0OT was 10 pg/ml (4 pg/col); the LLOQ for d5OT was 100 pg/ml (40 pg/col). For brain tissue, the LLOQ varied with the weight of the sample, therefore, the LLOQ for d0OT ranged from 0.00 to 0.03 pg; for d5OT, the LLOQ ranged from 0.07 to 0.44 pg. The upper limit of quantitation (ULOQ) for each assay in the method was 80 ng/ml, with accuracies between 80 and 120% at the LLOQ and ULOQ[41]. The method detection limit (MDL) was determined for each compound using a previously published method[42]. The MDL is defined as the minimum concentration of an analyte that can be measured and reported with 99% confidence that the analyte concentration is greater than zero. The MDL is determined from analysis of replicate standard injections in matrix at concentrations near the LLOQ to evaluate the uncertainty in the system. For determination of MDLs in this method, we performed five replicate injections, on 2 separate days, of the lowest concentration calibrator for each analyte, and calculated each MDL using an established formula[43]. An amount of analyte, equal to or greater than the MDL, is both detectable and distinguishable from the background with 99% confidence. The MDL for d0OT was 4 pg/ml and the MDL for d5OT was 38 pg/ml.

*Data analysis*: The LC–MS/MS data were analyzed using LabSolutions version 5.72 (Shimadzu). Target reference ion ratios were set according to the reference ion ratio of the highest standard. Default ion allowance for peak identification was 30%, relative to this target ratio. Linear regression with 1/C weighting was used for analysis of calibration curves.

**Blood and CSF sampling[44]**. Each monkey was initially anesthetized with 15 mg/kg (i.m.) ketamine. The animal was transferred to the ONPRC Surgical Services Unit, intubated, and maintained anesthetized with 1–1.5% isoflurane throughout tissue sampling procedures. Baseline blood samples were taken and d5OT was administered with an IV line in the saphenous vein or IN. Blood (5 ml) was drawn from the femoral region. CSF (750 μl/draw) was taken from the foramen magnum using a 25 g, 1 in. needle. ONPRC veterinarian staff placed an IV line and maintained isoflurane anesthesia for the 180 min of sample collection. Recovery took place in the housing cage under observation.

*Necropsy*: The necropsy procedure has been previously described[45,46]. The procedure is consistent with the recommendations of the American Veterinary Medical Association Guidelines on Euthanasia. The monkey was brought to a deep surgical plane of anesthesia with pentobarbital (30–50 mg/kg, IV). A thoracotomy followed by transcranial perfusion was performed, with pre-perfusion blood collected from the inferior vena cava. Perfusate buffer (ice-cold Krebs–Henseleit) consisted of (in mM) NaCl 137, Na2HPO4 6.4, Na2PO4 1.4, KCl 2.7, CaCl2 0.3, MgCl2 1, and glucose 5, with pH = 7.4. The amount of perfusate used during perfusion was ~1.5 L.

The brain was quickly removed following removal of the skull cap by saw, and then dissected into 4 mm slabs, while resting on cold aluminum plates (on wet ice). Regions of interest were rapidly cut and flash frozen.

**Data analysis**. *Brain tissue*: the concentration of d5OT in brain tissue in 12 brain regions for each DOSE–ROUTE condition: 80 IU IN, 80 IU IV, 40 IU IN is given in Table 1. Endogenous d0OT concentration in each of these regions is given in Table 2 (pg/mg) and 3B (nM) for each DOSE–ROUTE condition. Molarity was calculated using the sample weight for each brain region, the estimated density of rhesus brain tissue[47], the molecular weight of d0OT and d5OT (1007 and 1012 g/mol, respectively), and the mass of d0OT and d5OT in each sample. Volumes were adjusted to that of extracellular fluid volume, which was estimated to be 20% of brain volume[48].

*CSF and plasma analyses*: d5OT and d0OT concentrations in CSF and plasma outcome measures were examined for normality using the Kolmogorov–Smirnov test. Non-normal outcome measures were transformed (ln) and normalized values were used in subsequent analyses. Where outcome measures were not able to be normalized, nonparametric tests were used. Only measures of d5OT and d0OT that were ≥ the limit of quantification were used in the statistical analyses, as described below.

D5OT in plasma and CSF was entered into a linear Mixed Model, with within-subject factors of TIME (0, 60, or 120 min) and DOSE–ROUTE condition: 80 IU IN, 40 IU IN, 80 IU IV, or 40 IU IV. Analyses were conducted on d5OT levels in plasma and ln values in CSF, to determine if there was a main effect of DOSE–ROUTE, TIME, or DOSE–ROUTE × TIME interaction on the d5OT in the plasma and CSF.

Baseline difference in CSF d0OT levels was determined using a Linear Mixed Model with within-subject factors of DOSE–ROUTE condition: 80 IU IN, 40 IU IN, 80 IU IV, or 40 IU IV at Time = 0. Linear Mixed Model with within-subject factors of TIME (0, 60, or 120 min) and DOSE–ROUTE condition: 80 IU IN, 40 IU IN, 80 IU IV, or 40 IU IV was conducted on d0OT levels in CSF, to determine if there was an effect of DOSE–ROUTE, TIME, or DOSE–ROUTE × TIME interaction on d0OT in the CSF.

For significant interactions, planned post hoc analyses were concentration differences at each timepoint (60 and 120 min), as well as concentration differences between timepoints for each DOSE–ROUTE condition. All post hoc comparisons were Bonferroni corrected for multiple comparisons.

In order to determine whether there was a significant correlation between plasma and CSF d0OT, bivariate correlations were planned. Bivariate correlation was also conducted between plasma d5OT and CSF d0OT to determine whether plasma levels of d5OT varied significantly with endogenous CSF OT.

**Reporting summary**. Further information on research design is available in the Nature Research Reporting Summary linked to this article.

## Data availability

All raw data are available in Figs. 1–3 and Tables 1–4.

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

## Acknowledgements

This work was supported by (1) National Institutes of Health (NIH) intramural funding ZIA-AA000218 (Section on Clinical Psychoneuroendocrinology and Neuropsycho-pharmacology; PI: L.L.), jointly supported by the National Institute on Alcohol Abuse and Alcoholism (NIAAA) Division of Intramural Clinical and Biological Research and the National Institute on Drug Abuse (NIDA) Intramural Research Program (IRP); (2) NIH NIAAA R24 AA019431 grant (PI: K.A.G.); and (3) NIH Office of the Director P51 OD011092 grant (PI: Dr. P. Barr-Gielspe)

## Author contributions

M.R.L., K.A.G., and L.L. designed the study; T.A.S. conducted the experiments; S.W.B., A. V.K., A.J.W., and D.W.E. developed and conducted the mass spectrometry assays; M.R.L. analyzed the data; M.R.L., K.A.G., and L.L. wrote the paper; all authors contributed to and approved the paper.

## Competing interests

The authors declare no competing interests.

**Additional information**

