## [Peer Review File · Nature Communications]

Reviewers' comments:

Reviewer #1 (Remarks to the Author):

The study of Lee and co-authors, addressing permeability of neuropeptide oxytocin to the brain tissue of monkeys, is very timely and important for translational application of OT in human. Furthermore, the strength of the study is that it goes beyond oxytocin. With the intravenous condition as a control, the authors obtained two very valuable findings: 1) intranasal d5OT delivery bypasses the blood brain barrier; 2) intranasal drug delivery (at least of a small peptide, such as 9-aminoacid OT) holds promise for targeted drug delivery to specific brain sites that are in the trajectory of the olfactory and trigeminal nerves while limiting systemic exposure to the peptide.

The authors used a unique approach applying d5OT to detect it in plasma, CSF and brain tissue.

The work performed correctly, although the limited number of animals was employed. However, I have few concerns to discuss with the authors:

1. I guess, there is an error in Figure 1: there should be no label "INTRANASAL" in that figure.
2. What t=0 means? If this is the time point before d5OT injection, it should be clearly stated.
3. Would it be reasonable to add control group with saline injections?
4. It is surprising that no CSF d5 OT was detected at 60 or 120 minutes in CSF after IN administration. How the authors interpret these results in comparison to previous author's data and also work on human performed by Rene Hurlemann's group?
5. Why the authors did not measure OT in such important structures as the amygdala and insular? If you the authors did not measure OT or due to a detection, why these structures are listed it in the Table?

In general, more clear explanations of experimental design and results will straighten the manuscript. Indeed, the more details in the text will be profitable for easier reading and understanding the importance of author's findings.

Reviewer #2 (Remarks to the Author):

In this study, Lee et al administered intranasally as well as intravenously deuterated oxytocin (d5OT) to rhesus macaques and measured concentrations of exogenous and endogenous oxytocin (OT) in CSF, plasma, and most importantly 12 different brain regions. The authors reported that, after intranasal but not intravenous delivery, exogenous OT (d5OT) was found in a number of cortical and subcortical areas. In addition, intranasally delivered d5OT also persisted for a longer period of time in plasma. As the

authors correctly pointed out, despite the large number of acute OT administration studies in humans, in laboratory as well as clinical trials, little is known about the actual biochemical and neuronal mechanisms through which OT alters primate behavior. We commend the authors for making a very strong effort to bridge this critical gap in the OT literature. Nevertheless, we feel quite strongly that the scientific findings are not presented in a clear and effective manner, which limits its appeal to the broad readership of Nature Communications. We make several suggestions to improve the clarity of the manuscript.

Concerns with the clarity of the text:

Line 71-73: “Notwithstanding this action, clear evidence of the location and extent of brain parenchyma penetrance of OT after intranasal (IN) or intravenous (IV) administration remains unclear.”

--What “action”? And how can “clear evidence” be “unclear”?

Line 93 misses a closing bracket.

Line 93-95: “these regions were chosen as they are important in mediating social cognition in primate species and because they reflect regions associated with the mesolimbic social decision-making network in primate species.”

--I am not sure what the authors meant by saying areas such as the cerebellum and brainstem contribute to “social cognition” in primates, especially since the only relevant literature cited here is “Prefrontal Cortex and Social Cognition in Mouse and Man” (Bicks et al. 2015).

Line 98: “40 IU IN is the dose commonly used in human studies with OT.”

--This is not entirely true. Single-dose human studies usually use 20-40IU of OT, and most OT studies in rhesus monkeys were within this range (for example, 25 IU in Chang et al. 2013, Ebitz et al. 2013, Jiang & Platt 2018, 24 IU in Modi et al. 2014, 48 IU in Dal Monte et al. 2014). By this standard, 40 IU and especially 80 IU are on the higher end in terms of dosage for macaques. More importantly, since macaques weigh much less than adult humans, the IU/kg dose is quite high in the aforementioned studies. These caveats need to be properly addressed (Freeman et al. 2016).

Line 124-125: “This same device was used in similar previous nonhuman primate studies. Intranasal spray, anesthetized animals.”

--Here the authors cited the same paper (Dal Monte et al. 2014) twice. Additionally, Dal Monte et al. 2014 compared the effectiveness of intranasal spray and nebulization, but the authors failed to clarify that the method used in the current study was spray instead of nebulization. Furthermore, Dal Monte et al. reported that both delivery methods elevated OT level in CSF, while the changes in plasma OT were greater after nasal spray compared to nebulizer. Yet these findings (and other relevant studies such as Striepens et al. 2013, Modi et al. 2014) were never discussed. On a similar note, the CSF OT results presented here did not completely agree with the authors’ earlier paper (Lee et al. 2018), but this point was not sufficiently discussed either.

Study design and results:

My main concern is that too many statements seem redundant, self-contradictory, or simply call for further clarification.

Line 314-316: "Concentrations of d5OT were quantified only after IN administration, not IV administration; the brain regions where these were quantified were OFC, striatum, thalamus, and brainstem."

--I think the authors meant they "analyzed" all 12 brain areas following both administration routes, but they only "identified"/"detected" d5OT in OFC, striatum, thalamus, and brainstem in the IN condition.

Similarly, line 320-321: "D5OT in CSF was quantified only at time point 60...".

--I think the authors meant "detected" in this context, but please clarify.

Line 321: "There was no d5OT detected in plasma or CSF at T=0, as expected."

--Why was plasma d5OT discussed at all under the subtitle "CSF Concentrations of d5OT"?

Line 330-348: This section of text on plasma concentrations of d5OT constantly jumped back and forth comparing different doses, routes and time points with a series of wilcoxon signed rank tests. It is very unclear which specific hypotheses were being tested here and therefore difficult to gauge whether more appropriate statistical methods exist.

Line 347-348: "This difference indicates that IN delivery results in lower, but more sustained, plasma levels."

--The "lower" plasma OT concentration following IN delivery was not discussed in this section.

Line 359-360: "Bivariate correlation between d0OT in the CSF and d5OT concentration in the plasma was not significant: Pearson correlation=0.07, p=0.655."

The authors need to justify why this Pearson correlation was run and why it was the only correlation reported, and conveyed by a single sentence not belonging to any paragraph.

Line 448-450: "Decreasing concentrations, (~50%) over 2 hours, were observed in all 4 dose-route conditions, and this main effect of time does not support a feed forward effect of administered d5OT on endogenous levels."

--Why does this result not support a feed forward effect? Can the authors elaborate?

Line 463-464: "Another limitation is the 2 hour interval between d5OT administration and necropsy."

And

Line 467-472: "In addition, with the measurement of concomitant brain and CSF concentrations of d5OT, we can determine the extent to which IN administration results in brain concentrations of d5OT, in excess of that which crossed the blood brain barrier (i.e., IV route). This distinguishing feature is necessary, as the neurobehavioral effects of OT administered at peripheral sites, including IN, are most

likely centrally mediated, with OT signaling in the brain required for the behavioral effect.”

--These sentences seem to describe the strength of the current study but they lie within a paragraph dedicated to “limitations”.

Finally, I think the authors need to defend the study design more forcefully. It is quite understandable that nonhuman primate studies, especially neuroanatomical ones, will be constrained by the number of animals available. Nevertheless, it is important for researchers to define their hypotheses and complete the appropriate power analyses before conducting the experiment. For example, the authors acknowledged that, based on their prior study (Lee et al. 2018), they already knew some of the assay results could be “highly variable” across subjects (line 434). Despite this knowledge, they chose a mixed design with both males and females and 3 dose-route combinations, resulting in only two monkeys in each manipulation condition, and the only two male monkeys in the same condition (table 2). There are many potential problems with this design. First of all, the OT system is sexually dimorphic in a number of species (for example see Insel & Hulihan 1995; Dumais & Veenema 2016), and behavioral evidence suggests that male and female rhesus macaques respond to exogenous OT differently (Jiang & Platt 2018 a,b). If there were a sex-specific effect present in the current study, it would have been obscured by the dose-route effect.

More importantly, because of the large cross-subject variability (for example, in table 2, OT brain concentrations differed not only between the IN 40 and IN 80 conditions, but also frequently between the two subjects within the same condition), the current study is severely under-powered to uncover any cross-condition differences.

As the current manuscript stands, it fails to capture the interest of general readers because the specific questions, hypotheses, and motivation for each statistical test, and the greater impact of each finding, remain obscure. I think the manuscript would benefit significantly from extensive restructuring and rewriting to highlight the most important findings and implications for the field.

Reviewer #3 (Remarks to the Author):

In this paper, the authors examine accumulation of a labeled (deuterated) oxytocin in the brain following intranasal administration. This is a really important paper for several reasons, only one of which is the finding of brain penetration of exogenous oxytocin (d5OT). It also shows endogenous OT (d0OT) in areas of the brain that I am not aware of having been examined before in this manner in a primate (for example, the cerebellum).

There are a number of improvements which could be made in the paper, however, some substantive and some minor.

1) There should be details of the brain findings in the text of the results section. These findings are also

not discussed very thoroughly in the discussion and the authors don't make much of the novelty of their findings.

2) The tables are not very effective presentations of the data, and their legends are completely insufficient. I don't know if brain, CSF, and plasma are squished together this way because of figure limitations of the journal? But it makes it hard to read.

3) The figures also could be more attractive and use larger print for readers over 40!

4) For the d5OT, it appears that in some cases the compound was detectable at 40 IU IN but not at 80 IU IN (for example, in the striatum). This seems counter-intuitive...shouldn't a larger amount of d5OT show up in the same area in which a smaller amount could be detected? The authors should at least address what the reason for this might be.

5) There are a number of recent papers that are not cited here but are relevant, especially the following which posits a mechanism for OT to cross the blood-brain barrier:

<https://www.ncbi.nlm.nih.gov/pubmed/30820471>

But also the following which shows OTR binding in humans some of the areas examined:

<https://www.ncbi.nlm.nih.gov/pubmed/30514927>

And, since the authors are talking about neural OT in non-human primates and humans (and this paper includes data on OT fibers in rhesus monkeys);

<https://www.ncbi.nlm.nih.gov/pubmed/29797339>

Another important recent study on this topic:

<https://www.ncbi.nlm.nih.gov/pubmed/31238093>

6) The sentence beginning in line 309 is non-grammatical.

7) Line 261: should be baseline blood "samples"

Reviewer #4 (Remarks to the Author):

The work presents an important study on the administration of oxytocin via intranasal route (IN) versus IV in the rhesus monkey. As strong positives, the work is important, clinically relevant and is technically well performed.

The author's prior study weakens novelty. They have already demonstrated the approach, validated the use of this specific isotopic tracer and detection via LC/MS, and compared the effectiveness of IV and IN for CSF =. The current study builds and expands on the prior work as they looked at multiple (important) brain regions; not surprisingly, the peptide, once it reaches a brain region, lasts longer than in CSF.

An issue that is mentioned but not well addressed involves reproducibility of peptide delivery. The authors describe the large variability in the IN application. While the IV variability was 25-30% relative

variability, for IN, this increased to almost 100%. Given the lower variability in the IV, this does not appear to be a technical/measurement issue but an issue related to the mode of delivery. In such a case, when the peptide is applied by an expert, the amount delivered through IN does not appear to be well controlled. If the reviewer is understanding this, wouldn't this issue impact the clinical ability to deliver a specific amount of drug? Can the authors describe ways to reduce the individual to individual variability of peptides?

Reviewer #1 (Remarks to the Author):

The study of Lee and co-authors, addressing permeability of neuropeptide oxytocin to the brain tissue of monkeys, is very timely and important for translational application of OT in human. Furthermore, the strength of the study is that it goes beyond oxytocin. With the intravenous condition as a control, the authors obtained two very valuable findings: 1) intranasal d5OT delivery bypasses the blood brain barrier; 2) intranasal drug delivery (at least of a small peptide, such as 9-aminoacid OT) holds promise for targeted drug delivery to specific brain sites that are in the trajectory of the olfactory and trigeminal nerves while limiting systemic exposure to the peptide.

The authors used a unique approach applying d5OT to detect it in plasma, CSF and brain tissue.

The work performed correctly, although the limited number of animals was employed. However, I have few concerns to discuss with the authors:

1. I guess, there is an error in Figure 1: there should be no label “INTRANASAL” in that figure.

We thank the reviewer for pointing out this error and the “INTRANASAL” label as well as “INTRAVENOUS” has been removed from the figure; each plot is labelled with the IV dose of d5OT administered and this is also noted in the legend for Figure 1.

2. What $t=0$ means? If this is the time point before d5OT injection, it should be clearly stated.

Thank you for pointing this out. Yes, $T=0$ is just before d5OT administration. This is now clearly defined in the ‘Pharmacokinetic Analysis’ section of the manuscript and in the legends to Figures 1-3.

3. Would it be reasonable to add control group with saline injections?

As the main aim of this study was to determine the location and extent of *brain tissue* penetrance, this was a terminal study and as such, we devoted limited resources to 3 dose-route conditions: 80 and 40 IU IN, as well as 80IU IV. We do recognize that a saline condition would allow for a fully-factorial study design, however we are using labeled OT which permits us to differentiate unequivocally administered OT from endogenous OT. A saline condition would not significantly add to the achievement of this aim.

The decision of not including a saline condition was further guided by our previous monkey study (Lee et al., Mol Psychiatry, 2018) where we employed a control, saline condition, in addition to two experimental conditions: d5OT (80 IU) administered IN and IV. From this saline comparison, we determined that the time course of endogenous OT in the CSF after d5OT administration (IN or IV) was not different from the saline condition. Therefore, over this time period, there was not an effect of d5OT administration on endogenous OT CSF concentrations. We conclude from the present study that there is no difference between the 4 dose-route conditions with respect to the time course of endogenous OT in CSF and replicate the main effect of time to lower CSF OT concentrations.

4. It is surprising that no CSF d5 OT was detected at 60 or 120 minutes in CSF after IN administration. How the authors interpret these results in comparison to previous author's data and also work on human performed by Rene Hurlemann's group?

In our previous study (Lee et al., Mol Psychiatry 2018), after IN administration, CSF d5OT concentration at 60 minutes was quantified in 4 monkeys as follows: 11.28, 15.99, 67.65 and 221.62 pg/ml. In this previous study, the limit of quantification (LOQ) for d5OT was 10 pg/ml. The LOQ of d5OT in the present study was 100 pg/ml and the method detection limit (MDL) for d5OT was 38 pg/ml. Half of the d5OT concentrations in CSF at 60 minutes, in the previous study after IN administration, were below the MDL of the present assay. Therefore, it is not surprising that we did not detect d5OT in CSF at 60 or 120 minutes in the present study, given the considerable variability in d5OT concentrations at across monkeys at each timepoint.

In a between-subjects study in humans, Rene Hurlemann's group (Striepens et al., Sci Rep 2013) administered intranasally, 24 IU of OT (not labelled) and demonstrated a significant increase in CSF OT concentration to approximately 30 pg/ml compared to placebo at 75 minutes post-administration. In this study, OT was quantified with a radioimmunoassay (Riagnosis, Munich, Germany) whose sensitivity is 0.1 pg/ml. Nevertheless, this concentration (30 pg/ml) was also below the MDL for the assay employed in the present study. Also, in the human study at 60 minutes, there was no between-group (administered OT vs placebo) differences in CSF OT concentrations, indicating no apparent change in CSF OT concentration as a consequence of intranasal IN administration. However, comparison of this study to the present one is difficult, as the analysis in the former study was limited to one timepoint between groups and the assay was based on a different methodology.

5. Why the authors did not measure OT in such important structures as the amygdala and insular? If you the authors did not measure OT or due to a detection, why these structures are listed it in the Table?

We did attempt to measure d5OT and d0OT concentrations in the amygdala and insula in 5 of 6 monkeys after administration of 80 IU IN and 80 IU IV d5OT, in addition to one monkey in the 40 IU IN d5OT condition. Please see Tables 2 and 3 for notation of conditions where the assay was not employed. In one monkey (Monkey ID #1) who received d5OT 40 IU IN, we did not measure brain tissue from the amygdala or insula, due to technical difficulties.

In general, more clear explanations of experimental design and results will straighten the manuscript. Indeed, the more details in the text will be profitable for easier reading and understanding the importance of author's findings.

Thank you, the manuscript has been edited throughout for clarity in description of design and results.

Reviewer #2 (Remarks to the Author):

In this study, Lee et al administered intranasally as well as intravenously deuterated oxytocin (d5OT) to rhesus macaques and measured concentrations of exogenous and endogenous oxytocin (OT) in CSF, plasma, and most importantly 12 different brain regions. The authors reported that, after intranasal but not intravenous delivery, exogenous OT (d5OT) was found in a number of cortical and subcortical areas. In addition, intranasally delivered d5OT also persisted for a longer period of time in plasma. As the authors correctly pointed out, despite the large number of acute OT administration studies in humans, in laboratory as well as clinical trials, little is known about the actual biochemical and neuronal mechanisms through which OT alters primate behavior. We commend the authors for making a very strong effort to bridge this critical gap in the OT literature. Nevertheless, we feel quite strongly that the scientific findings are not presented in a clear and effective manner, which limits its appeal to the broad readership of Nature Communications. We make several suggestions to improve the clarity of the manuscript.

Concerns with the clarity of the text:

Line 71-73: “Notwithstanding this action, clear evidence of the location and extent of brain parenchyma penetrance of OT after intranasal (IN) or intravenous (IV) administration remains unclear.”

--What “action”? And how can “clear evidence” be “unclear”?

This sentence in the first paragraph of the Introduction has been rewritten, p 3.

Line 93 misses a closing bracket.

This has been corrected.

Line 93-95: “these regions were chosen as they are important in mediating social cognition in primate species and because they reflect regions associated with the mesolimbic social decision-making network in primate species.”

--I am not sure what the authors meant by saying areas such as the cerebellum and brainstem contribute to “social cognition” in primates, especially since the only relevant literature cited here is “Prefrontal Cortex and Social Cognition in Mouse and Man” (Bicks et al. 2015).

Thank you for pointing this out. We also cite Grinevich et al., Biological Psychiatry, 2016, who note that brainstem regions in primate species may be important in social behavior. We have added a reference (Van Overwalle et al., Medical Hypotheses, 2019) noting the important role of the cerebellum in social cognition in primate species.

Line 98: “40 IU IN is the dose commonly used in human studies with OT.”

--This is not entirely true. Single-dose human studies usually use 20-40IU of OT, and most OT studies in rhesus monkeys were within this range (for example, 25 IU in Chang et al. 2013, Ebitz et al. 2013, Jiang & Platt 2018, 24 IU in Modi et al. 2014, 48 IU in Dal Monte et al. 2014). By this standard, 40 IU and especially 80 IU are on the higher end in terms of dosage for macaques. More importantly, since macaques weigh much less than adult humans, the IU/kg dose is quite

high in the aforementioned studies. These caveats need to be properly addressed (Freeman et al. 2016).

We have now noted that 40 IU IN is in the upper range of doses used in human studies administering oxytocin intranasally, please see p. 4.

(2)

Line 124-125: “This same device was used in similar previous nonhuman primate studies. Intranasal spray, anesthetized animals.”

--Here the authors cited the same paper (Dal Monte et al. 2014) twice. Additionally, Dal Monte et al. 2014 compared the effectiveness of intranasal spray and nebulization, but the authors failed to clarify that the method used in the current study was spray instead of nebulization. Furthermore, Dal Monte et al. reported that both delivery methods elevated OT level in CSF, while the changes in plasma OT were greater after nasal spray compared to nebulizer. Yet these findings (and other relevant studies such as Striepens et al. 2013, Modi et al. 2014) were never discussed. On a similar note, the CSF OT results presented here did not completely agree with the authors' earlier paper (Lee et al. 2018), but this point was not sufficiently discussed either.

We now draw these distinctions between methods of delivery and note these other studies. Citation of Dal Monte et al.'s paper twice was an error that we have corrected. Discussion on the present work and on our earlier work (Lee et al., Mol Psychiatry 2018) has been expanded.

Study design and results:

My main concern is that too many statements seem redundant, self-contradictory, or simply call for further clarification.

Line 314-316: “Concentrations of d5OT were quantified only after IN administration, not IV administration; the brain regions where these were quantified were OFC, striatum, thalamus, and brainstem.”

--I think the authors meant they “analyzed” all 12 brain areas following both administration routes, but they only “identified”/“detected” d5OT in OFC, striatum, thalamus, and brainstem in the IN condition.

Thank you for this suggestion. We have made the distinction between analysis, detection and quantification in the Results section.

Similarly, line 320-321: “D5OT in CSF was quantified only at time point 60...”.

--I think the authors meant “detected” in this context, but please clarify.

Yes, this section has been clarified with respect to detection and quantification as above.

Line 321: “There was no d5OT detected in plasma or CSF at T=0, as expected.”

--Why was plasma d5OT discussed at all under the subtitle “CSF Concentrations of d5OT”?

Mention of plasma d5OT has been moved to the ‘Plasma Concentrations of d5OT’ section.

Line 330-348: This section of text on plasma concentrations of d5OT constantly jumped back and forth comparing different doses, routes and time points with a series of wilcoxon signed rank tests. It is very unclear which specific hypotheses were being tested here and therefore difficult to gauge whether more appropriate statistical methods exist.

Only pairwise analyses were conducted within the same route or same dose at a single timepoint or between timepoints. This is clarified on p 16 of the manuscript.

Line 347-348: “This difference indicates that IN delivery results in lower, but more sustained, plasma levels.”

--The “lower” plasma OT concentration following IN delivery was not discussed in this section.

This sentence has been clarified to note that after IN administration, there are lower concentrations compared to IV at 60 minutes and these plasma concentrations are sustained.

Line 359-360: “Bivariate correlation between d0OT in the CSF and d5OT concentration in the plasma was not significant: Pearson correlation=0.07, p=0.655.”

The authors need to justify why this Pearson correlation was run and why it was the only correlation reported, and conveyed by a single sentence not belonging to any paragraph.

Thank you for pointing this out. In order to determine whether there was a significant correlation between plasma and CSF d0OT, bivariate correlations were planned. Bivariate correlation was conducted between plasma d5OT and CSF d0OT to determine whether plasma levels of d5OT varied significantly with endogenous CSF OT. This is now explained on pp 12-13. In the Results section, p 15, we now state that Bivariate correlation between d0OT in plasma and CSF was not possible as endogenous OT in plasma was not able to be quantified.

Line 448-450: “Decreasing concentrations, (~50%) over 2 hours, were observed in all 4 dose-route conditions, and this main effect of time does not support a feed forward effect of administered d5OT on endogenous levels.”

--Why does this result not support a feed forward effect? Can the authors elaborate?

Thank you for this comment, we have clarified on p 18 that if a feed forward effect were present, we would have expected increasing concentrations of d0CSF over time after d5OT administration.

Line 463-464: “Another limitation is the 2 hour interval between d5OT administration and necropsy.”

And

Line 467-472: “In addition, with the measurement of concomitant brain and CSF concentrations of d5OT, we can determine the extent to which IN administration results in brain concentrations

of d5OT, in excess of that which crossed the blood brain barrier (i.e., IV route). This distinguishing feature is necessary, as the neurobehavioral effects of OT administered at peripheral sites, including IN, are most likely centrally mediated, with OT signaling in the brain required for the behavioral effect.”

--These sentences seem to describe the strength of the current study but they lie within a paragraph dedicated to “limitations”.

We have reworded this paragraph to clarify the point about the limitation of the 2 hour interval between d5OT administration and perfusion.

Finally, I think the authors need to defend the study design more forcefully. It is quite understandable that nonhuman primate studies, especially neuroanatomical ones, will be constrained by the number of animals available. Nevertheless, it important for researchers to define their hypotheses and complete the appropriate power analyses before conducting the experiment. For example, the authors acknowledged that, based on their prior study (Lee et al. 2018), they already knew some of the assay results could be “highly variable” across subjects (line 434). Despite this knowledge, they chose a mixed design with both males and females and 3 dose-route combinations, resulting in only two monkeys in each manipulation condition, and the only two male monkeys in the same condition (table 2). There are many potential problems with this design. First of all, the OT system is sexually dimorphic in a number of species (for example see Insel & Hulihan 1995; Dumais & Veenema 2016), and behavioral evidence suggests that male and female rhesus macaques respond to exogenous OT differently (Jiang & Platt 2018 a,b). If there were a sex-specific effect present in the current study, it would have been obscured by the dose-route effect.

More importantly, because of the large cross-subject variability (for example, in table 2, OT brain concentrations differed not only between the IN 40 and IN 80 conditions, but also frequently between the two subjects within the same condition), the current study is severely under-powered to uncover any cross-condition differences.

As the current manuscript stands, it fails to capture the interest of general readers because the specific questions, hypotheses, and motivation for each statistical test, and the greater impact of each finding, remain obscure. I think the manuscript would benefit significantly from extensive restructuring and rewriting to highlight the most important findings and implications for the field.

We have revised the Introduction to emphasize that the main aim of this terminal study in nonhuman primates was to determine if and where d5OT appeared in the brain after administration by the IN route, and whether there was evidence that IN administration bypassed the blood brain barrier. As pointed out in the Introduction, establishing whether IN administration of OT results in brain penetrance is a critically important first step in the interpretation of the results of clinical studies with IN OT, and in progress toward development of IN OT as a therapeutic medication for numerous neuropsychiatric disorders.

The additional aims and hypotheses were ordered, expanded and clarified. We also have added to the limitation section of the Discussion that we were not powered to detect sex differences in endogenous brain OT concentrations, or detect a response in endogenous OT as a consequence of administered OT. On that note, it is, however, important to keep in mind that addressing sex differences was not an aim of this study, and we appreciate that these results should be replaced in larger samples that will be powered adequately to detect potential sex differences. Nonetheless, we did include both male and female monkeys in this study, a decision consistent with the recent NIH policy stating that NIH expects researchers to study both male and female vertebrate animals and humans.

Reviewer #3 (Remarks to the Author):

In this paper, the authors examine accumulation of a labeled (deuterated) oxytocin in the brain following intranasal administration. This is a really important paper for several reasons, only one of which is the finding of brain penetration of exogenous oxytocin (d5OT). It also shows endogenous OT (d0OT) in areas of the brain that I am not aware of having been examined before in this manner in a primate (for example, the cerebellum).

We thank the reviewer for these positive comments.

There are a number of improvements which could be made in the paper, however, some substantive and some minor.

1) There should be details of the brain findings in the text of the results section. These findings are also not discussed very thoroughly in the discussion and the authors don't make much of the novelty of their findings.

Please see the Results section where this is now discussed and the Discussion section where the novelty is discussed in the first paragraph.

2) The tables are not very effective presentations of the data, and their legends are completely insufficient. I don't know if brain, CSF, and plasma are squished together this way because of figure limitations of the journal? But it makes it hard to read.

Tables 2 and 3 have been revised, noting where concentrations were not detected and specifying this in the legend. The CSF and Plasma concentrations were removed from Table 2 and are now in the Results section of the manuscript, pp 14-16. Table 2B has been removed and nanomolar (nM) concentrations have been moved beside concentrations expressed as pg/mg. nM concentrations remain for endogenous OT brain concentrations in Table 3B

3) The figures also could be more attractive and use larger print for readers over 40!

The Figures have been enlarged and improved with color coding to distinguish, for each monkey, the time course of endogenous and exogenous OT concentrations in plasma and CSF.

4) For the d5OT, it appears that in some cases the compound was detectable at 40 IU IN but not at 80 IU IN (for example, in the striatum). This seems counter-intuitive...shouldn't a larger amount of d5OT show up in the same area in which a smaller amount could be detected? The authors should at least address what the reason for this might be.

Discussion of this point is now in the paragraph on limitations in the Discussion section, p 21.

5) There are a number of recent papers that are not cited here but are relevant, especially the following which posits a mechanism for OT to cross the blood-brain barrier:

<https://www.ncbi.nlm.nih.gov/pubmed/30820471>

But also the following which shows OTR binding in humans some of the areas examined:

<https://www.ncbi.nlm.nih.gov/pubmed/30514927>

And, since the authors are talking about neural OT in non-human primates and humans (and this paper includes data on OT fibers in rhesus monkeys);

<https://www.ncbi.nlm.nih.gov/pubmed/29797339>

Another important recent study on this topic:

<https://www.ncbi.nlm.nih.gov/pubmed/31238093>

We thank the reviewer for suggesting these additions and these references have been cited and discussed in the manuscript.

6) The sentence beginning in line 309 is non-grammatical.

7) Line 261: should be baseline blood "samples"

These errors have been corrected.

Reviewer #4 (Remarks to the Author):

The work presents an important study on the administration of oxytocin via intranasal route (IN) versus IV in the rhesus monkey. As strong positives, the work is important, clinically relevant and is technically well performed.

The author's prior study weakens novelty. They have already demonstrated the approach, validated the use of this specific isotopic tracer and detection via LC/MS, and compared the effectiveness of IV and IN for CSF =. The current study builds and expands on the prior work as they looked at multiple (important) brain regions; not surprisingly, the peptide, once it reaches a

brain region, lasts longer than in CSF.

An issue that is mentioned but not well addressed involves reproducibility of peptide delivery. The authors describe the large variability in the IN application. While the IV variability was 25-30% relative variability, for IN, this increased to almost 100%. Given the lower variability in the IV, this does not appear to be a technical/measurement issue but an issue related to the mode of delivery. In such a case, when the peptide is applied by an expert, the amount delivered through IN does not appear to be well controlled. If the reviewer is understanding this, wouldn't this issue impact the clinical ability to deliver a specific amount of drug? Can the authors describe ways to reduce the individual to individual variability of peptides?

This point is mentioned now on p 20 of the manuscript. We also would like to thank the reviewer for these positive comments.

Reviewer #1 (Remarks to the Author):

The manuscript has been significantly improved and deserves publication.

Reviewer #2 (Remarks to the Author):

The authors have addressed many of our concerns in this revised manuscript. The data are important to present since this is the first measurement and comparison of IV and intranasal OT penetrance in the primate brain.

That said, several concerns with the manuscript remain:

1. The finding of no difference in CSF OT concentration between IN OT administration and IV administration does not align with findings from previous studies (e.g. Chang et al. 2014; Dal Monte et al. 2014; Striepens et al. 2013). The authors need to more fully defend this result against the findings of prior studies. Is this a limitation of their method or the use of deuterated OT? Deuteration is known to affect the pharmacokinetics of drugs.
2. The figures are a mess. In figures 1 and 2 the 40 IU plots are missing from the figures. In figure 3, the y-axes are all on different scales, apparently chosen arbitrarily, making comparison difficult. For all figures in which the reader is invited to compare conditions, the data should be plotted on the same y-axes.
3. The text is still awkward in places, with some tortured construction, misplaced commas, missing spaces, etc. I don't have time to go through line by line, but this paper could use some proof-reading.

Reviewer #4 (Remarks to the Author):

The authors responded to the reviewer comments in detail. The comments involved several technical questions, presentation issues and finally some advice on additional citations and organization to aid the clarity of the work. The authors well addressed the technical issues and unclear presentations. There are no further issues. In addition, the authors have added clarifying statements and citations as requested. Overall, the manuscript as been improved. As all four reviewers stated, the work is important and exciting, and the revised manuscript does a much better job at conveying the importance of the work.

Reviewers' comments:

Reviewer #1 (Remarks to the Author):

The manuscript has been significantly improved and deserves publication.

We thank the reviewer for the positive feedback on this revised manuscript.

Reviewer #2 (Remarks to the Author):

The authors have addressed many of our concerns in this revised manuscript. The data are important to present since this is the first measurement and comparison of IV and intranasal OT penetrance in the primate brain.

That said, several concerns with the manuscript remain:

1. The finding of no difference in CSF OT concentration between IN OT administration and IV administration does not align with findings from previous studies (e.g. Chang et al. 2014; Dal Monte et al. 2014; Striepens et al. 2013). The authors need to more fully defend this result against the findings of prior studies. Is this a limitation of their method or the use of deuterated OT? Deuteration is known to affect the pharmacokinetics of drugs.

These 3 studies (Chang et al. 2014; Dal Monte et al. 2014; Striepens et al. 2013) differed from our study in two important ways: 1) they did not administer labelled OT and 2) the quantification of OT was not by mass spectrometry. Therefore, administered and endogenous OT could not be quantified separately. As such, the methods employed in the present study were an advance which allowed us to measure the effect of administered OT on endogenous concentrations in the CSF.

In the present study, in CSF, we did not quantify d5OT after IN administration. Therefore, it was not possible to compare CSF d5OT concentrations between IV and IN administration. Consistent with our previous study (Lee et al., Molecular Psychiatry 2018), we found no effect of d5OT administration on endogenous OT CSF.

In plasma, we found no difference at 60 minutes or at 120 minutes between plasma concentrations of d5OT after IN compared to IV administration. This is in contrast to our previous study and is likely due to the lower sensitivity of the assay in the present study, variability between monkeys, missing data and the non-normal data necessitating use of a nonparametric statistical test. These points are now noted in the Discussion, p. 19.

Deuteration platforms are used to stabilize small molecules and prolong their half-life (1). However, this approach stabilizes molecules that are metabolized via oxidation (1). Metabolism of oxytocin is via hydrolysis so the peptide's pharmacokinetics are unlikely to be altered by deuteration. This is now noted in the Discussion, p. 20.

2. The figures are a mess. In figures 1 and 2 the 40 IU plots are missing from the figures. In figure 3, the y-axes are all on different scales, apparently chosen arbitrarily, making comparison

difficult. For all figures in which the reader is invited to compare conditions, the data should be plotted on the same y-axes.

We have corrected the error in Figures 1 and 2, as well as made the y axis scale uniform across each of the 4 plots in Figure 3.

3. The text is still awkward in places, with some tortured construction, misplaced commas, missing spaces, etc. I don't have time to go through line by line, but this paper could use some proof-reading.

We have proof-read and edited the manuscript for clarity and style.

Reviewer #4 (Remarks to the Author):

The authors responded to the reviewer comments in detail. The comments involved several technical questions, presentation issues and finally some advice on additional citations and organization to aid the clarity of the work. The authors well addressed the technical issues and unclear presentations. There are no further issues. In addition, the authors have added clarifying statements and citations as requested. Overall, the manuscript as been improved. As all four reviewers stated, the work is important and exciting, and the revised manuscript does a much better job at conveying the importance of the work.

Thank you for this positive feedback.

1. Tung R. The development of deuterium-containing drugs. *Innovations in Pharmaceutical Technology*. 2010;32(32):24-8.

Reviewer #2 (Remarks to the Author):

The authors have responded to my concerns. In particular, the figures are much better. The paper reaches my threshold for publication.

REVIEWERS' COMMENTS

Reviewer #2 (remarks to author):

The authors have responded to my concerns. In particular, the figures are much better. The paper reaches my threshold for publication.

We thank the reviewer for the helpful suggestions to improve this manuscript.